# A distribution model for *Glossina brevipalpis* and *Glossina austeni* in Southern Mozambique, Eswatini and South Africa for enhanced area-wide integrated pest management approaches

**Chantel J. de Beer**[1,2]\*, **Ahmadou H. Dicko**[3], **Jerome Ntshangase**[2], **Percy Moyaba**[2], **Moeti O. Taioe**[2], **Fernando C. Mulandane**[4], **Luis Neves**[4,5], **Sihle Mdluli**[6], **Laure Guerrini**[7,8], **Jérémy Bouyer**[1,7,9], **Marc J. B. Vreysen**[1], **Gert J. Venter**[2,5]

1 Joint FAO/IAEA Centre of Nuclear Techniques in Food and Agriculture, Insect Pest Control Laboratory, Department of Nuclear Sciences and Applications, International Atomic Energy Agency, Vienna, Austria, 2 Epidemiology, Parasites & Vectors, Agricultural Research Council—Onderstepoort Veterinary Research (ARC-OVR), Onderstepoort, South Africa, 3 STATS4D, Dakar, Senegal, 4 Biotechnology Centre, Eduardo Mondlane University, Maputo, Mozambique, 5 Vectors and Vector Borne Diseases Research Program, Department of Veterinary Tropical Diseases, Faculty of Veterinary Science, University of Pretoria, Onderstepoort, South Africa, 6 Epidemiology Unit, Department of Veterinary Services, Manzini, Eswatini, 7 UMR ASTRE (Animal, Health, Territories, Risks and Ecosystems), CIRAD, INRA, Université de Montpellier, Montpellier, France, 8 RP-PCP, UMR ASTRE, Harare, Zimbabwe, 9 UMR INTERTRYP, Univ Montpellier, CIRAD, IRD, Montpellier, France

\* c.de-beer@iaea.org

## Abstract

### Background

*Glossina austeni* and *Glossina brevipalpis* (Diptera: Glossinidae) are the sole cyclical vectors of African trypanosomes in South Africa, Eswatini and southern Mozambique. These populations represent the southernmost distribution of tsetse flies on the African continent. Accurate knowledge of infested areas is a prerequisite to develop and implement efficient and cost-effective control strategies, and distribution models may reduce large-scale, extensive entomological surveys that are time consuming and expensive. The objective was to develop a MaxEnt species distribution model and habitat suitability maps for the southern tsetse belt of South Africa, Eswatini and southern Mozambique.

### Methodology/Principal findings

The present study used existing entomological survey data of *G. austeni* and *G. brevipalpis* to develop a MaxEnt species distribution model and habitat suitability maps. Distribution models and a checkerboard analysis indicated an overlapping presence of the two species and the most suitable habitat for both species were protected areas and the coastal strip in KwaZulu-Natal Province, South Africa and Maputo Province, Mozambique. The predicted presence extents, to a small degree, into communal farming areas adjacent to the protected areas and coastline, especially in the Matutuíne District of Mozambique. The quality of the

**Data Availability Statement:** Vegetation classes map and models described in the paper, are

available in this link https://dataverse.harvard.edu/dataset.xhtml?persistentId=doi:10.7910/DVN/PA7U7L.

**Funding:** CdB and GV received funding for this work from the Joint Food and Agriculture Organization of the United Nations (FAO)/International Atomic Energy Agency (IAEA) Centre of Nuclear Techniques in Food and Agriculture and the IAEA's Department of Technical Cooperation. AD received funding from the IAEA's Department of Technical Cooperation for model developing. Field collections in South Africa were funded by the Department of Science and Technology. LG and JB received funding from the GeosAf project (www.rp-pcp.org/projects/) implemented by the ACP Group of States. This publication has been produced with the assistance of the European Union or that of the funders. The contents of this publication are the sole responsibility of authors and can in no way be taken to reflect the views of the European Union. The funders had no role in study design, data collection and analysis, decision to publish, or preparation of the manuscript. This work was also conducted within the framework of the Research Platform « Production and Conservation in Partnership » (www.rp-pcp.org).

**Competing interests:** The authors have declared that no competing interests exist.

MaxEnt model was assessed using an independent data set and indicated good performance with high predictive power (AUC > 0.80 for both species).

## Conclusions/Significance

The models indicated that cattle density, land surface temperature and protected areas, in relation with vegetation are the main factors contributing to the distribution of the two tsetse species in the area. Changes in the climate, agricultural practices and land-use have had a significant and rapid impact on tsetse abundance in the area. The model predicted low habitat suitability in the Gaza and Inhambane Provinces of Mozambique, i.e., the area north of the Matutuíne District. This might indicate that the southern tsetse population is isolated from the main tsetse belt in the north of Mozambique. The updated distribution models will be useful for planning tsetse and trypanosomosis interventions in the area.

## Author summary

The two tsetse species transmitting nagana in South Africa, Eswatini and southern Mozambique represent the southernmost distribution of this genus on the African continent. Distribution models were developed to support tsetse control. These models indicated that the main factors contributing to tsetse distribution in the area are the presence of host animals, variation in climate and vegetation mostly observed in protected areas, agricultural practises and land-use also had a significant and rapid impact on tsetse abundance in the area. Application of the model to areas north of the southern distribution predict a low presence of suitable habitats in the Gaza and Inhambane Provinces of Mozambique, thereby indicating that this southern population is geographically isolated from the main tsetse belt starting in the north of Mozambique.

## Introduction

Tsetse flies (Diptera: Glossinidae) are considered the sole cyclical vectors of African trypanosomes and are reported to occur in about 10 million km$^2$ in sub-Saharan Africa [1]. The trypanosome parasites cause Human African Trypanosomosis (HAT) or sleeping sickness in humans and African Animal Trypanosomosis (AAT) or nagana in livestock. Both diseases have a substantial negative effect on agricultural development and economic growth in sub-Saharan Africa [2,3]. HAT is a fatal disease if left untreated and, although absent in southern Africa, occurs regularly in some regions of sub-Saharan Africa with 70 million people at risk of becoming infected in 36 countries [4].

Tsetse flies are restricted to sub-Saharan Africa [5] and they have been sampled as far south as the north-eastern parts of KwaZulu-Natal Province (KZN) of South Africa (Latitude S28°31'13.44"). This southern population extends into the neighbouring Maputo Province (MP) of Mozambique [6–11]. Of the 31 described tsetse fly species and subspecies, only two species are found as far south as South Africa, i.e., *Glossina brevipalpis* Newstead belonging to the *Fusca* (forest) species group and *Glossina austeni* Newstead belonging to the *Morsitans* (savannah) species group [1,7,8].

In addition to several species of wildlife, both species feed on cattle [5] and are involved in the transmission of two pathogenic protozoa, *Trypanosoma congolense* and *Trypanosoma*

*vivax* [12–14] that cause the debilitating disease AAT in livestock. *T. congolense* is the most abundant species in South Africa [13] and limited vector competence studies indicated that *G. austeni* was the more competent vector for *T. congolense* in this area [14,15].

Tsetse infested areas in KZN are mainly used for communal farming and inhabited by 426 000 humans, 130 000 small ruminants and 360 000 cattle [16]. Livestock production and agricultural development is severely hindered by these flies as vectors of trypanosomes which causes considerable stress to the farmers, not only in KZN but in the entire distribution area of the two species [14]. The *G. brevipalpis* belt extends from Ethiopia in north-eastern Africa southwards to Somalia, Uganda, Kenya, Rwanda, Burundi and Tanzania [10]. In southern Africa the presence of *G. brevipalpis* extends from Malawi, Zambia, Zimbabwe, Mozambique to the north-eastern parts of KZN [5]. *G. austeni* is found in East Africa from Somalia in the north, extending southwards into Kenya, Tanzania, Zimbabwe, Eswatini, Mozambique and the north-eastern parts of KZN [5,10]. *G. austeni* was also present on Unguja Island of Zanzibar, Tanzania, but the population was eradicated in 1997, after implementing an area-wide integrated pest management (AW-IPM) campaign that included the sterile insect technique (SIT) [17].

The most southern distribution of these two species is shared by Mozambique (Matutuíne District), Eswatini (Mlawula Nature Reserve) and South Africa (north-eastern KZN). Localised surveys and available distribution prediction models for South Africa showed that, on a microecological scale, *G. brevipalpis* and *G. austeni* are confined to pockets of dense vegetation in north-eastern KZN [18–21]. This apparent patchy distribution, in addition to possible low migration potential [7,18], may have resulted in the development of localised genetically isolated population pockets [22]. Preliminary studies, using molecular and phenetic (geometric morphometrics) markers, however, suggested an absence of significant barriers to gene flow within the pockets of this southern population [23]. These data provided evidence that *G. brevipalpis* and *G. austeni* populations in southern Mozambique, Eswatini and South Africa can be considered homogenous and that an eradication strategy that is limited to one country will not be sustainable because of potential reinvasion from uncontrolled neighbouring areas [23]. However, this southern tsetse population might be geographically isolated from the main tsetse belt that starts approximately 500 km north of Matutuíne District, south of the Save River in central Mozambique [6–9,11,24]. If confirmed, this will offer an opportunity to sustainably create a tsetse-free zone in southern Africa.

The current strategy in South Africa to manage nagana relies on long-term vector suppression using live-bait technologies and the *ad hoc* treatment of cattle with trypanocidal drugs. For the live bait technology, the chemical amitraz ($C_{19}H_{23}N_3$) that has normally been used in the existing network of dipping tanks for the management of tick-borne diseases, was replaced with a wide-spectrum insecticide, i.e., the pyrethroid cyhalothrin ($C_{23}H_{19}ClF_3NO_3$) that is more effective against dipteran flies [25]. This cattle-dipping regime for suppressing the tsetse populations has been adopted since 2015 (personal communication Dr. L. Ntantiso, Department of Agriculture, KwaZulu-Natal, South Africa). Although pyrethroids as a dipping agent are effective for the management of both ticks and tsetse flies [26], ticks can develop pyrethroid resistance with a reduction in its efficiency as an acaricide as previously reported in the area [26,27]. This livestock dipping strategy can also only be effective if a large proportion of the tsetse population feeds on domestic rather than wild animals [28]. Managing AAT with pyrethroids in dipping tanks can therefore only be a temporary solution at best. The establishment of a tsetse-free zone in southern Africa would be a more sustainable solution to the nagana problem. A potential sustainable solution to eradicate *G. brevipalpis* and *G. austeni* from north-eastern KZN was proposed in 2007 [7] and was based on an AW-IPM strategy that includes an SIT component. In the absence of significant barriers to gene flow between the

populations of MP, Eswatini, and KZN [23], the proposed strategy [7] will have to be adapted to target sequentially the populations in all three countries.

Determining the precise and potential geographic distribution and abundance of the targeted tsetse population will be vital for the success of an AW-IPM strategy. The size of the area that needs to be treated will directly affect the outcome, sustainability and cost of any control campaign. Because of the logistical and funding constraints involved in implementing surveys over such a large rural area that extends over three countries, models that predict the distribution, presence and even abundance of the tsetse populations would facilitate the development and implementation of these surveys and make them much more focussed and cost-effective [29].

The first probability of presence model developed for South Africa was based on tsetse fly sampling data collected with sticky XT traps between 1993 and 1999 [19,30]. This model incorporated climate and environmental variables and predicted a more extensive geographical distribution for both *G. brevipalpis* and *G. austeni* than was indicated by the sticky trap data [19], suggesting that the model may have overestimated the distribution [30]. Although this model was updated and refined by including data from tsetse sampled with the more effective H-trap between 2005 and 2007, there were still areas where the predicted probability of presence could be improved. Incorporating data on cattle density, human population, agricultural intensity and detailed vegetation biomes were suggested to improve model fit [20]. Furthermore, the earlier models did not include southern Mozambique and Eswatini. The objective of the study was to develop a MaxEnt species distribution model and habitat suitability maps for the southern tsetse belt of South Africa, southern Mozambique and Eswatini using available entomological survey data of *G. austeni* and *G. brevipalpis*. The updated models may be able to predict to what extent changes in land use and agricultural practises may potentially influence tsetse abundance and the occurrence of AAT.

## Materials and methods

### Study area

In South Africa the envisaged tsetse infested area (16 000 km²) stretches from around 10 km south of the Mfolozi River in the south, for approximately 190 km, to the border of Mozambique in the north, and from the Indian Ocean coast in the east for 80 km to the Hluhluwe-iMfolozi Park in the west (Fig 1) [7,8]. The infested area extends into the Matutuine District (8 500 km²) (MP) of Mozambique, the northern limit being the Boane and Namaacha Districts of Mozambique [6,11]. In the east, it borders with the Indian Ocean and in the west with Eswatini. In Eswatini, the Mlawula Nature Reserve was surveyed. The reserve is located west of the Lebombo Mountains (elevation 776 m), an 800 km-long narrow range of mountains that stretch from Hluhluwe in KZN in the south to Punda Maria in the Limpopo Province in South Africa in the north parallel with the Mozambique border.

Farming systems are predominantly subsistence farming with numerous communal farms interspersed with several protected areas consisting of provincial and private game parks and reserves. This includes the iSimangaliso Wetland Park which was listed as South Africa's first World Heritage Site in December 1999. These areas contain a wide variety of game animals consisting of large numbers of bigger mammals, small primates, rodents as well as birds that are potential hosts for tsetse. The South African target area contains several state forests, mostly pine and eucalyptus plantations, and sugarcane farms.

The area has a large range of land cover relevant to the presence or absence of *G. brevipalpis* and *G. austeni* such as coastal forests, savannah and agricultural areas. The climate is subtropical, except the mountainous "highveld" area in Eswatini that has a temperate climate.

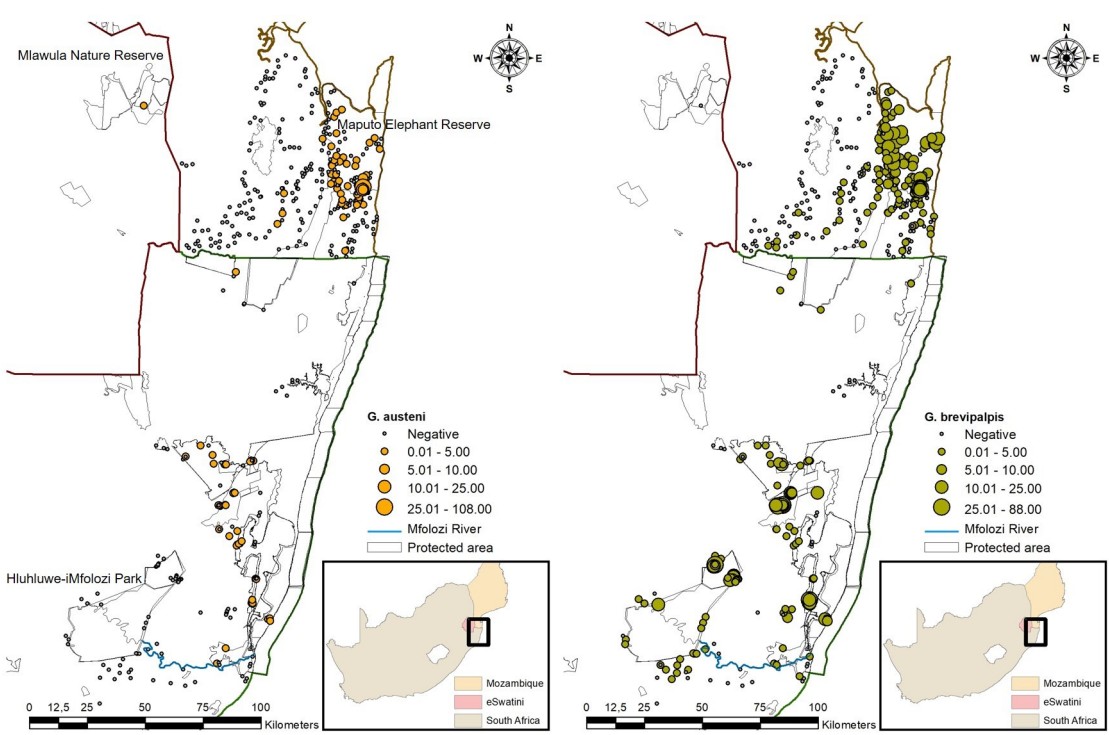

**Fig 1. Apparent density of *Glossina austeni* and *Glossina brevipalpis* collected between 2009 and 2019 from Maputo Province (MP), Mozambique, Eswatini and north-eastern KwaZulu-Natal Province, South Africa (https://dataverse.harvard.edu/dataset.xhtml?persistentId=doi:10.7910/DVN/PA7U7L).**

## Tsetse data collection

**South Africa.** Data obtained from routine entomological surveys using H-traps, carried out between April and May 2012, April and June 2015, March 2016 and February 2017 and October 2018 and June 2019, were used for model development (Fig 1). Trapping sites were selected in areas of known tsetse presence or with a high probability of tsetse presence as predicted by a previous presence model [19,30]. To enhance the trapping of *G. brevipalpis*, each trap was baited with the odours 1-octen-3-ol and 4-methylphenol at a 1:8 ratio and released at 4.4 mg/h and 7.6 mg/h, respectively [31]. The chemicals were dispensed from seven heat-sealed sachets (7 cm x 9 cm) made of low-density polyethylene sleeves (wall thickness 150 μm) placed near the entrance of each trap. A 300 mL brown glass bottle with a 6 mm hole in the lid dispensed acetone at a rate of ca. 350 mg/h and was placed close to the trap [18,21]. In total 160 H-traps were deployed for a minimum of 19 and a maximum of 244 trapping days. Considering the number of traps deployed per site and the number of trapping days it is envisaged that the influence on tsetse densities was minimal. Flies were collected from the traps and the traps serviced (e.g., clearing of vegetation, odour replacement, replace the traps when the colours are faded) every 14 days. The traps contained two plastic bottles for fly collection. The bottles contained a 20% ethanol to which an antiseptic, Savlon (Johnson & Johnson (Pty) Ltd., Rattray Road, East London, South Africa) was added to preserve the sampled flies as well as to prevent ant and spider predation. The collected tsetse flies were identified morphologically to species level and sexed. The number of each species collected over this period was counted and results expressed as apparent density (AD), i.e., the number of flies per trap per day.

**Mozambique.** In the Matutuíne District (MP) of Mozambique, entomological surveys were carried out annually between 2009 to 2013 and in June-July 2019 (Fig 1). Tsetse flies were

sampled with 283 odour-baited H-traps following the protocol as described above for South Africa, with the exception that the traps were deployed for a minimum of three days and a maximum of 14 days.

**Eswatini.** In Eswatini collections were made in the Mlawula Nature Reserve from April 2019 to June 2019 with 10 odour-baited H-traps (Fig 1). The Mlawula Nature Reserve was the only area were tsetse flies were trapped in a country-wide survey carried out from 12 April to 7 May 2008 [24]. The sampling protocol was the same as that used in South Africa and Mozambique.

## Data analysis

**Tsetse occurrence and density.** The data collected during these independent entomological surveys, were correlated with vegetation classes developed for the area and incorporated in models to predict suitable habitat for potential tsetse distribution. The statistical software R version 3.6.2. using RStudio Desktop version 1.2.5033 [32] was used for data analysis. A co-occurrence analysis, consisting of a binary presence-absence matrix, was carried out to assess if the distribution of *G. austeni* and *G. brevipalpis* throughout the study area was segregated, aggregated or random. A Checkerboard score (C-score) [28] that is measuring aggregation or segregation intensity (checkerboardness) was calculated. To evaluate the tsetse relative abundance, a one-way analysis of variance (ANOVA) was used to compare the mean tsetse fly AD between sites. The data were not normally distributed and nonparametric Kruskal-Wallis tests were used as a post test. Additionally, Dunn's multiple comparison tests were used where P value < 0.05.

**Spatial filtering and generation of pseudo-absence.** All datasets were spatially filtered to reduce the spatial correlation in the modelling process. One record was kept per pixel (500 m) and any absence data within a buffer of 2 km around a presence data was removed from the analysis. A probabilistic model was applied to remove non-significant absence from the analysis (5% level) [33,34]. Absence data were kept for validation and some background (pseudo-absence) was generated randomly in the grid according to the kernel based on sampling effort. While higher sampling effort increases the pseudo-absence generated, areas less sampled will have less. The kernel was built using spatial point pattern analysis incorporating all available trap information from the study area. Subsequently, one pseudo-absence per raster cell (approx. 500 m x 500 m) was selected. Finally, all pseudo-absences into a 2 km radius buffer around presence data were removed, taking into consideration the average flight range of these species. To reduce the sampling bias these pseudo-absence data were generated by taking into account the environment and the range of the efficiency of the H-traps to sample both species. A model based on a multidimensional nonparametric kernel [35] was developed in areas of high sampling effort to correct for sampling bias. The models were fitted with *G. austeni* and *G. brevipalpis* presence data collected with H-traps from the three countries. Additionally, the models were validated using presence and absence data for both species.

**Remote sensing data.** Time series of high spatial resolution (100 m to 1000 m) remote sensing data were downloaded, cleaned and summarized to build relevant covariates. Ten years of Moderate-resolution Imaging Spectroradiometer (MODIS) products (January 2010 to December 2019) were used.

A detailed vegetation classification map of the tsetse distribution area in South Africa, Eswatini and southern Mozambique was derived from four Landsat 8 images (resolution 30*30m, June to July 2013) and field surveys conducted in July 2013. The field records comprised 1092 photographs taken at locations within the known tsetse distribution area of South Africa, Eswatini and southern Mozambique. The photographs were used to identify 11 main

types of vegetation based on the international nomenclature that were considered relevant for the presence or absence of *G. brevipalpis* and *G. austeni* (Fig 2), i.e., savannah woodland, herbaceous savannah, shrub savannah, dense dry forest, gallery forest, tree plantations, crops (agricultural areas), urban areas, swamps, water bodies and bare ground were the mainland-cover classes. MODIS products and indices were selected to capture the complexity of tsetse habitat preference, in particular, thermal and vegetation indices. MODIS products were acquired from the NASA Earth Observing System data server, the human population density layer from the Worldpop project [34], cattle density from the FAO Database [36], FAO

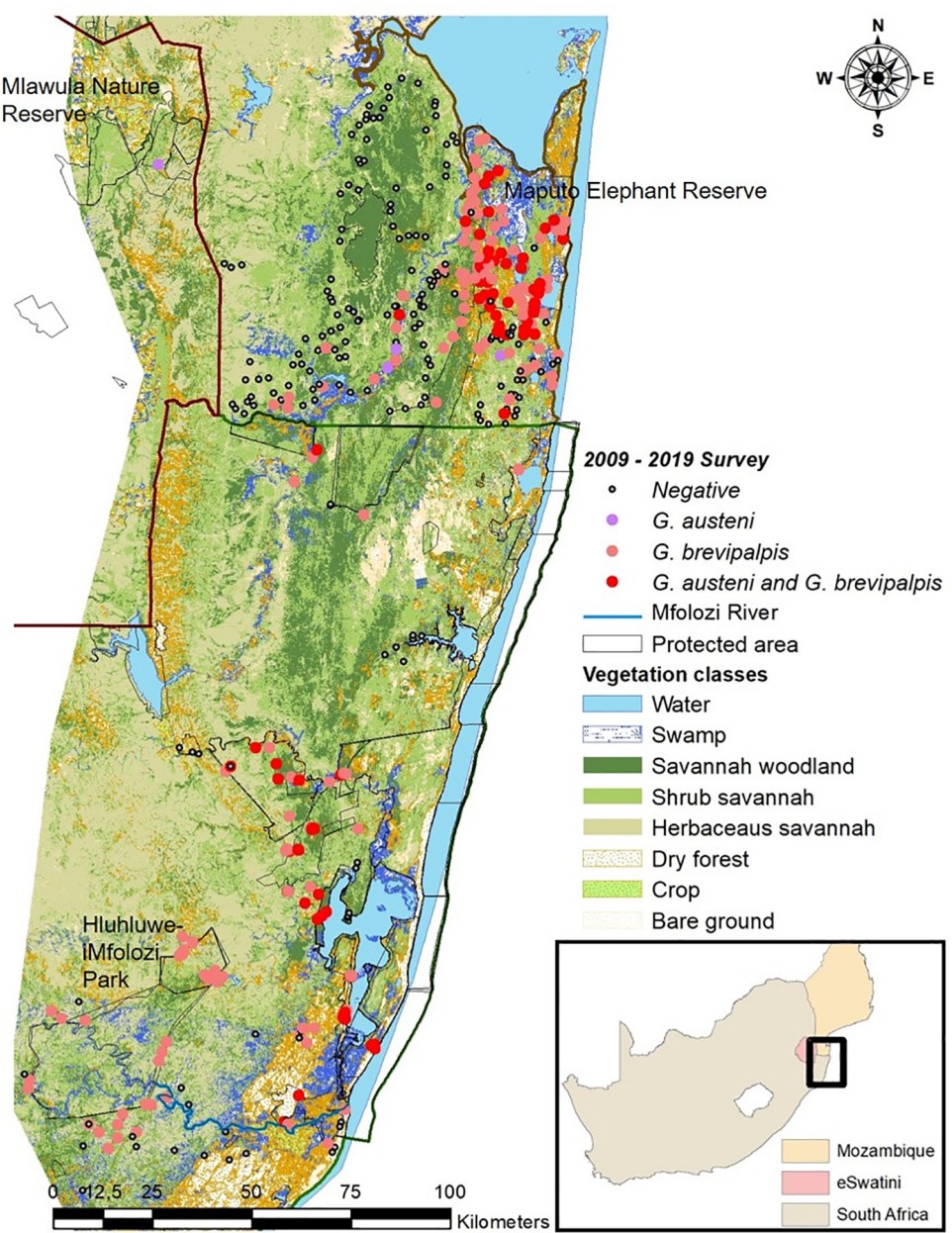

**Fig 2. Vegetation classes map with *Glossina austeni* and *Glossina brevipalpis* presence and absence collected between 2009 and 2019 (https://dataverse.harvard.edu/dataset.xhtml?persistentId=doi:10.7910/DVN/PA7U7L).**

**Table 1. Variables derived from remote sensing data used in the model.**

| Variable Name | | Type | Product | Spatial Resolution (m) | Temporal Resolution (days) | Source |
|---|---|---|---|---|---|---|
| Normalized Difference Vegetation Index | NDVI | Vegetation | MOD13A1/MYD13A1 | 500 x 500 | 16 | MODIS |
| Middle Infra-Red | MIR | Vegetation | MOD13A1/MYD13A1 | 500 x 500 | 16 | MODIS |
| Enhanced Vegetation Index | EVI | Vegetation | MOD13A1/MYD13A1 | 500 x 500 | 16 | MODIS |
| Tree cover | treecov | Vegetation | MOD44B | 250 x 250 | 365 | MODIS |
| Day land surface temperatures | DLST | Thermal | MOD11A2/MYD11A2 | 1000 x 1000 | 8 | MODIS |
| Night Land surface temperatures | NLST | Thermal | MOD13A2/MYD11A2 | 1000 x 1000 | 8 | MODIS |
| Human Population density | POP | | | 100 x 100 | | WorldPop |
| Cattle Density | Cattle | | Livestock Gridded of the World | 1000 x 1000 | | FAO |
| Slope | Slope | Topographic | | 500 x 500 | | SRTM |
| Aspect | Aspect | Topographic | | 500 x 500 | | SRTM |

livestock gridded of the world project [37], and layers of protected areas from the World Database on Protected Areas [38].

For each variable, data were pre-processed and cleaned by re-interpolating the data at a spatial resolution of 500 m using the nearest neighbour method. A MODIS QA (pixel quality) mask was applied to remove poor quality pixels. Thermal data were filtered using the boxplot algorithm to reduce the effect of outliers [39]. Summary statistics such as mean, maximum, minimum and range were computed for the variables Normalized Difference Vegetation Index (NDVI), Middle Infra-Red (MIR), Enhanced Vegetation Index (EVI), Tree Cover (treecov), Day Land Surface Temperatures (DLST) and Night Land Surface Temperatures (NLST) (Table 1).

**Model selection and validation.** Ecological Niche Factor Analysis (ENFA), a variant of factor analysis, was used to explore and model species ecological niche [40]. The environmental space used by the species was compared with the available environmental space using two indicators i.e., marginality and specialization. Marginality was used to measure niche central position. It captures the dimension in the ecological space in which the average conditions where the species accrues differs from the global conditions. A large marginality value implies that the conditions where the species was found were "far" from the global environmental conditions. In contrast, specialization measures the spread and usage of the ecological space along dimensions of niche use. The higher this value, the narrower the space used by the species. Consequently, the species niche can be summarized by an index for marginality and specialization and represented on a factor map within the biplot framework.

The Maximum Entropy ecological niche model (MaxEnt), one of most common species distribution models [41], was used to predict the potential distribution of the two species. It uses a machine learning method based on the information theory concept of maximum entropy [42]. MaxEnt fits a species distribution by contrasting the environmental condition where the species is present, and the environment characterized by some pseudo-absence data also called background. The logistic output from this method is a suitability index that ranges between 0 (low suitable habitat) and 1 (high suitable habitat). This output was used to create suitability maps for both species.

Predictors used in these models were chosen according to the ecology of the two species involved. Twenty-four bioclimatic variables based on remote sensing data were built to model the distribution of these two species. For each of the remote sensing derived index, we computed some measures of position (mean, min, max) and some measures of spread (range,

coefficient of variability). Multiple models were fitted varying the background of the model to assess the effect of pseudo-absence and use model averaging [43]. The effect of pseudo-absences was assessed by repeating the process to generate them several times and monitor variability and stability in model quality metrics. Furthermore, the predictive quality of the different MaxEnt models was assessed with a subset of the entomological survey data that was excluded from the analysis. Uncertainty was measured using the coefficient of variation from the predictions. The main metric used to check predictive power of the different models was the Area Under the ROC Curve (AUC). A likelihood-based metrics was also analysed to assess the importance and contribution of each variable.

Based on the predictive quality of the developed MaxEnt models, and keeping the degree of uncertainty in mind, the potential distribution of the two species was determined by extrapolation of the model to adjoining areas outside the collection limits.

## Results

### Tsetse apparent density and vegetation classification

The 485 H-traps deployed in 45 sites in KZN, Eswatini and MP, collected 61 316 *G. brevipalpis* (AD = 4.32 flies/trap/day) and 1378 *G. austeni* (AD = 0.10 flies/trap/day) between August 2009 and June 2019 (Fig 1). While both species were collected in KZN and MP, only *G. austeni* was trapped in Eswatini. The two species had strongly aggregated distributions in the entire study area (C-sore = 0.014, R < 0.001) (Fig 1).

The mean AD (4.10 ± 6.4) of *G. brevipalpis* in KZN was significantly higher (P < 0.01) than that (2.14 ± 5.4) in MP. However, the lower mean AD (0.08 ± 0.2) of *G. austeni* in KZN was not significantly different from that (0.90 ± 6.9) of MP. The mean AD of *G. austeni* collected in Eswatini was 0.34 ± 0.2.

In KZN, *G. brevipalpis* was most abundant in dense dry forest vegetation ($\bar{x}$ AD of 10.47 ± 7.7) (Figs 1 and 2) followed by shrub savannah ($\bar{x}$ AD of 6.46 ± 7.9) and savannah woodland ($\bar{x}$ AD of 5.37 ± 7.3). The lowest population densities of *G. brevipalpis* were found in the swamp areas ($\bar{x}$ AD of 1.26 ± 0.03). In MP, *G. brevipalpis* was most abundant in shrub savannah ($\bar{x}$ AD of 5.70 ± 6.7) followed by savannah woodland ($\bar{x}$ AD of 4.0 ± 9.0), dense dry forest ($\bar{x}$ AD of 3.71 ± 5.9) and swamp areas ($\bar{x}$ AD of 3.10 ± 5.19). Overall, from the entire collection area, significantly (P < 0.01) more *G. brevipalpis* were collected from dense dry forest ($\bar{x}$ AD of 6.39 ± 7.4) as compared with shrub savannah ($\bar{x}$ AD of 5.60 ± 7.1), savannah woodland ($\bar{x}$ AD of 4.64 ± 8.3), swamp ($\bar{x}$ AD of 2.70 ± 4.6) and herbaceous savannah ($\bar{x}$ AD of 2.20 ± 2.3).

In KZN, most *G. austeni* were sampled in savannah woodland ($\bar{x}$ AD of 0.20 ± 0.3), followed by dense dry forest ($\bar{x}$ AD of 0.15 ± 0.2), shrub savannah ($\bar{x}$ AD of 0.03 ± 0.1) and herbaceous savannah ($\bar{x}$ AD of 0.004 ± 0.01) (Figs 1 and 2). This species was not sampled in the swamp areas. In MP, the highest population densities of *G. austeni* were found in dense dry forest ($\bar{x}$ AD of 3.24 ± 15.2) followed by savannah woodland ($\bar{x}$ AD of 1.41 ± 4.7), shrub savannah ($\bar{x}$ AD of 0.57 ± 1.9), herbaceous savannah ($\bar{x}$ AD of 0.2 ±0.4) and the swamp areas ($\bar{x}$ AD of 0.05 ± 0.1). In Eswatini, *G. austeni* was only sampled in savannah woodland areas with a mean AD of 0.34 ± 0.2.

Considering the entire collection area, similar to *G. brevipalpis*, significantly (P < 0.01) more *G. austeni* were collected in dense dry forest ($\bar{x}$ AD of 2.02 ± 11.9) as compared with savannah woodland ($\bar{x}$ AD of 0.87 ± 3.6), shrub savannah ($\bar{x}$ AD of 0.36 ± 1.5), herbaceous savannah ($\bar{x}$ AD of 0.11 ± 0.3) and swamp areas ($\bar{x}$ AD of 0.04 ± 0.1).

In KZN and MP, the relative abundance of both species was significantly (P < 0.01) greater in protected areas as compared with the areas surrounding these protected areas, i.e., for *G.*

*austeni* a mean AD of 0.13 ± 0.2 was observed inside protected areas in KZN versus a mean AD of 0.02 ± 0.2 outside, and in MP a mean AD of 1.7 ± 9.6 inside versus a mean AD of 0.005 ± 0.03 outside (Fig 1). The mean ADs of both species decreased significantly (P < 0.01) with distance from a protected area in both countries. The mean AD of *G. austeni* collected in a 5 km buffer zone around the protected areas were 0.03 ± 0.2 and 0.01 ± 0.1 in KZN and MP, respectively. Beyond this 5 km buffer zone, the mean AD (0.0003 ± 0.001) of *G. austeni* collected in KZN was ten times lower as compared with the mean AD (0.003 ± 0.01) in MP. The trend was similar for *G. brevipalpis* with mean AD's of 1.01 ± 1.8 in KZN and 0.65 ± 1.3 in MP within the 5 km buffer zone around the protected areas and a reduction further away, i.e., 0.67 ± 1.9 in KZN and 0.02 ± 0.1 in MP.

## Maximum entropy ecological niche model (MaxEnt)

Results of the ENFA showed that *G. austeni* occurrence was positively associated with vegetation indices (EVI range (5.1%) and tree cover mean (2.2%)) as well as presence of protected areas (Fig 3). The same positive association with vegetation indices and presence of protected areas was observed for *G. brevipalpis* (Fig 3). Human population and cattle densities and most of the temperature indices showed a negative association with occurrence of both species, except the minimum night land surface temperature which was positively correlated with occurrence of *G. brevipalpis*. Slope and aspect were positively related to *G. brevipalpis* presence but negatively with *G. austeni*. Mean night land surface temperature influenced the habitat for the two species. Minimum MIR for *G. austeni* and mean MIR for *G. brevipalpis* accounted for most of the variance and fell outside the cloud of average conditions available in the study area (Fig 3).

For *G. austeni*, the variable cattle density contributed 24.8% towards model fitting, followed by the range of day land surface temperatures (16.7%) and presence of protected areas (13.9%). For *G. brevipalpis*, the variable cattle density contributed 42.6% towards model fitting, followed by mean night land surface temperature (12.9%) and presence of protected areas (8.9%) (Fig 4).

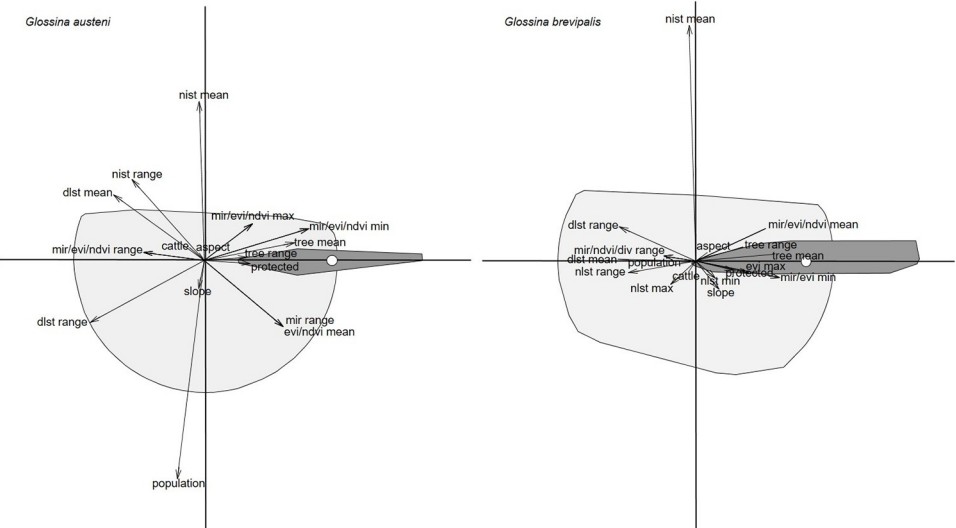

**Fig 3. The ecological niche factor analysis plan.** Light grey polygon shows the overall environmental conditions available in the study area, dark grey polygon shows environmental conditions where *Glossina austeni* and *Glossina brevipalpis* were observed, and the white circles corresponds to the barycentre of its distribution.

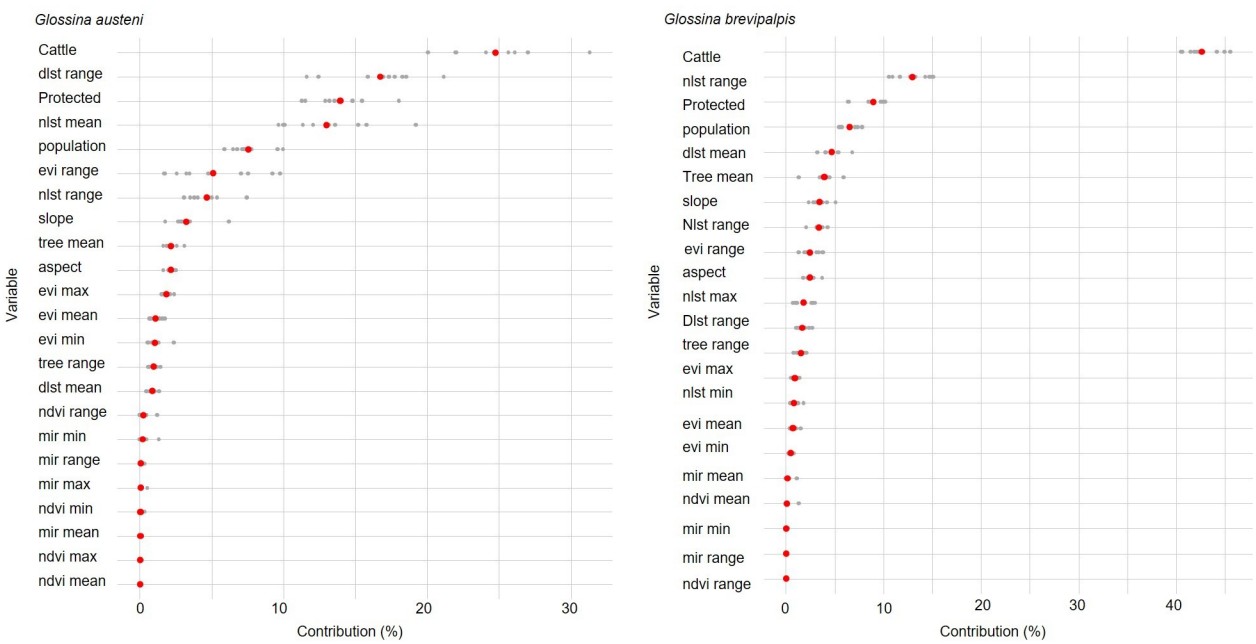

**Fig 4. Contribution of variables to the suitability index by decreasing importance for *Glossina austeni* and *Glossina brevipalpis*.** The 95% confidence interval is indicated in red and individual values in grey.

For the Maxent models the mean AUC as assessed with the independent data was 0.88 (range 0.87 to 0.89) and 0.84 (range 0.81 to 0.85) for *G. austeni* and *G. brevipalpis*, respectively (Fig 5). The uncertainty grid for the habitat suitability index model (Mass analyses) indicated that the uncertainty in the predictions was low, except in areas west of the known tsetse infested zone (S1 Fig). When the model was extended northwards to the Gaza and Inhambane Provinces of Mozambique the uncertainty increased. However, the prediction values along the coastline in the Inhambane Province retained relative high certainty. The distribution of the two species was similar with high suitability areas in and along the protected areas in all three countries and the coastal areas of north-eastern KZN and MP.

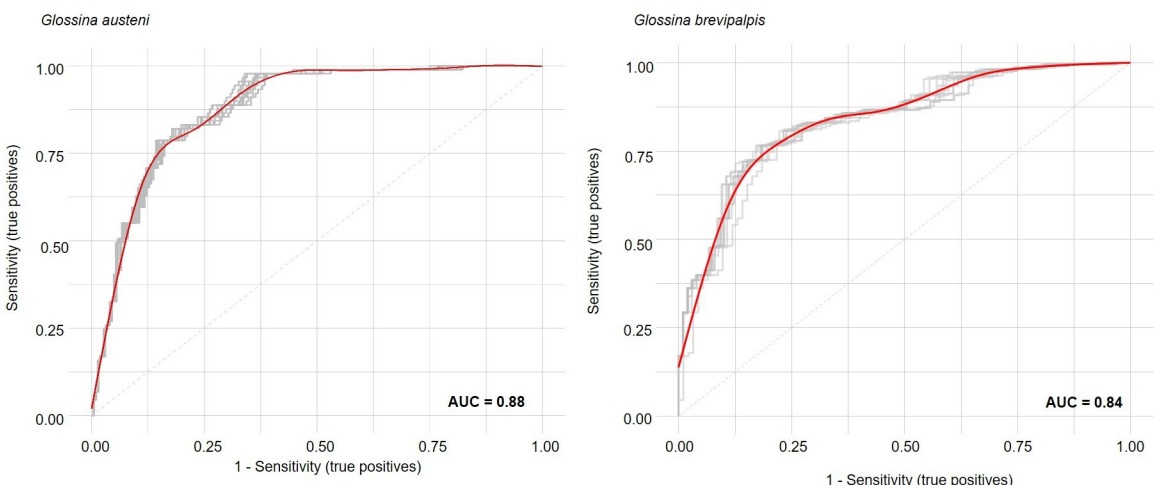

**Fig 5.** Area under the curve for the average MaxEnt model (in red) and the 10 sub models (in grey) for *Glossina austeni* and *Glossina brevipalpis*.

Suitable habitat for *G. austeni* was predicted mainly along the coastline and inside protected areas of north-eastern KZN and MP (Fig 6). The predicted suitable habitat extended to a small degree into the communal farming areas close to the protected areas and coastline. In Eswatini the suitable predicated habitat was also linked to protected areas along the eastern border of Mozambique. There was also a small suitable area in the central part of Eswatini.

Overall *G. brevipalpis* displayed a wider distribution than *G. austeni* (Fig 1). The suitable habitat for *G. brevipalpis* was highly linked to protected areas but extended to a greater degree into adjoining communal cattle farming areas as compared with *G. austeni*. This extension was more pronounced in the MP than in KZN. Suitable habitat for *G. brevipalpis* was also predicted over a narrow strip at the border between Eswatini and Mozambique, and, like *G. austeni*, in the central part of Eswatini, but to a larger extent. In north-eastern KZN, a suitable band was predicted in the northern part in the communal farming area along the coast and border of the iSimangaliso Wetland Park. The largest area of predicted suitable habitat for *G. brevipalpis* in the communal farming area was west and south of Hluhluwe-imfolozi Park. The area of high probability for *G. brevipalpis* extended south along the coast far beyond its known historical distribution limits.

The model was extended for both species (Fig 7) to include the Gaza and Inhambane Provinces of Mozambique. The model indicated that large parts of this area were unsuitable for both species, but the habitat was more suitable for *G. brevipalpis* than for *G. austeni*. Areas of high suitability were mostly found along the coast and in a small area of the Zinave National Park close to the Save River.

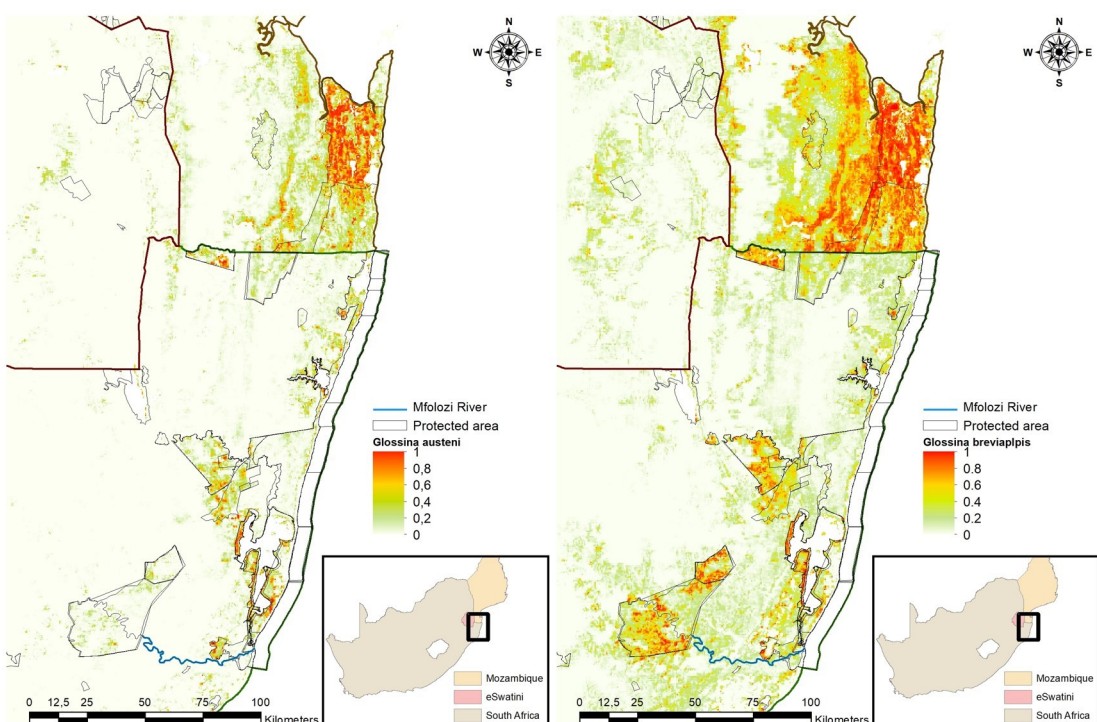

**Fig 6. The mean habitat suitability index predicted by a MaxEnt model for *Glossina austeni* and *Glossina brevipalpis* for Maputo Province (MP), Mozambique, Eswatini and north-eastern KwaZulu-Natal Province, South Africa (https://dataverse. harvard.edu/dataset.xhtml?persistentId=doi:10.7910/DVN/PA7U7L).**

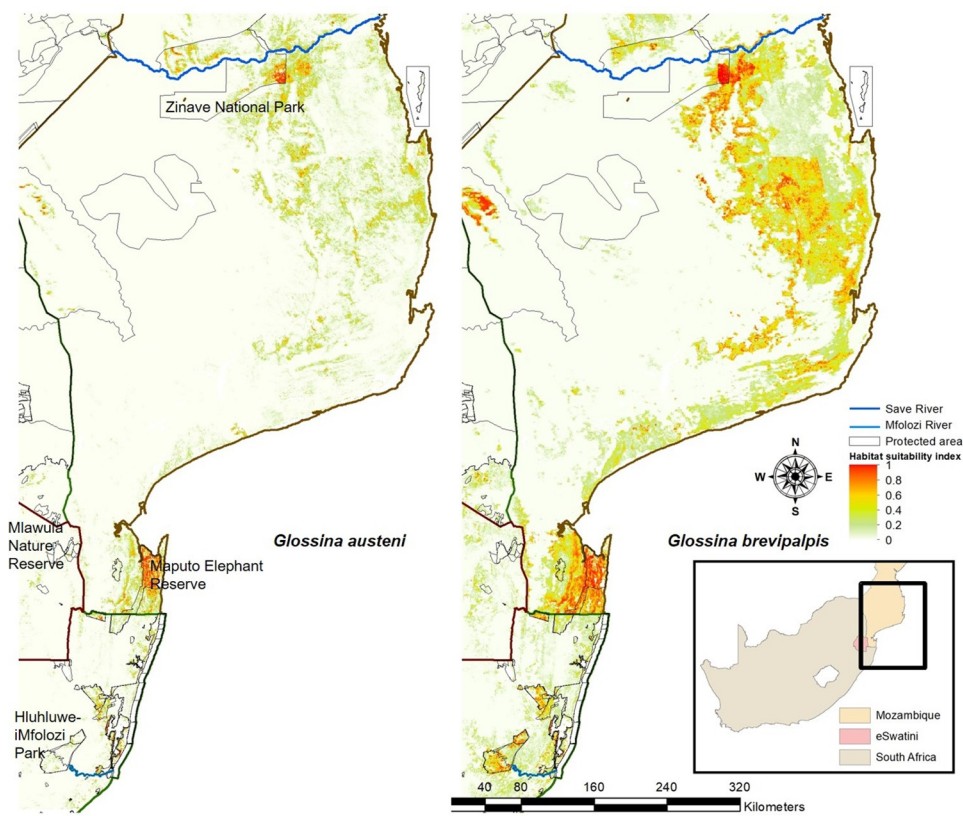

**Fig 7. Extended mean habitat suitability index predicted by a MaxEnt model for *Glossina austeni* and *Glossina brevipalpis* in Mozambique, Eswatini and South Africa (https://dataverse.harvard.edu/dataset.xhtml?persistentId=doi:10.7910/DVN/PA7U7L).**

## Discussion

A sound understanding of the potential distribution of targeted tsetse fly species will be essential for the successful implementation of any AW-IPM programme that include an SIT component, e.g., it will establish the area over which sterilized males need to be released in programmes. The development of habitat suitability maps, and an assessment of the factors that regulate the presence or absence of the targeted species, will therefore be advantageous, if not essential, to ensure that these programmes will be efficient and cost-effective. Reliable prediction maps will enable more efficient planning of entomological surveys and selection of appropriate trapping sites. The present study used existing entomological survey data to develop habitat suitability maps for *G. austeni* and *G. brevipalpis* using statistical methods whilst incorporating what is known of the ecology of both species, e.g., host and habitat preferences. The methodology followed to develop these prediction maps was similar to the one used in the AW eradication programme of *Glossina palpalis gambiensis* Vanderplank in the Niayes of Senegal [29,41]. The MaxEnt approach was used as previous studies in West Africa [41] indicated that it better predicted suitable landscapes and tsetse presence as compared with a presence-absence regularized logistic regression model. The MaxEnt model was develop based on ADs as obtained with odour baited H-traps for both species. Although the H-trap was specifically developed for the collection of these species [44] the differential efficacy of this trap remains unknown for the two species. In order to develop better informed and robust models, systematic sampling within study area is still the most efficient way to improve this

model. In agreement with previous entomological surveys in the affected area in north-eastern KZN, climate, vegetation and presence of protected areas and cattle were the key regulators that determined the presence and abundance of *G. brevipalpis* and *G. austeni* [8,16,18,21].

Previous surveys and distribution prediction models showed that the *G. austeni* and *G. brevipalpis* populations were, on a micro-ecological scale, mainly confined to pockets of dense vegetation in north-eastern KZN [18,19,21,30]. A strong relationship between tsetse relative abundance and vegetation type was observed in the current study. High relative abundance of *G. austeni* and *G. brevipalpis* was strongly associated with savannah woodland and dense dry forests, respectively. Low relative abundance of *G. austeni* and *G. brevipalpis* was associated with swamp and crop areas, respectively. This association was evidenced by the positive association between tree cover and predicted suitability for both species. These observations highlight the regulatory role that the presence of suitable vegetation may play in predicting the presence or absence of these two tsetse fly species in an area and the success of bush clearing as an earlier method for tsetse control [45].

The model predicted suitable habitat for both species along the coast of north-eastern KZN and MP (Fig 6). Previous studies conducted in north-eastern KZN suggested a relationship between the relative abundance of the tsetse populations and the range of temperature variation [8]. In accordance with model predications the relative abundance of both species was higher at coastal sites compared to sites in the interior, and this can be related to the variation in the average temperature and relative humidity, which are less pronounced at the coast as compared with the interior. The range in day land surface temperatures contributed as much as 16.7% towards model fitting for *G. austeni*. For *G. brevipalpis* the contribution of the range in day land surface temperature (1.6%) was of less importance than the mean night land surface temperatures (12.9%). This relationship was also reflected in the prediction model, as there was a negative association between probability of presence and an increase in the range of day surface temperatures.

The model indicated that the predicted presence of both species is associated with the presence of protected game areas in all three countries. Protected areas contributed 13.9% and 8.9% towards model fitting for *G. austeni* and *G. brevipalpis*, respectively. These protected areas harbour the preferred hosts of *G. brevipalpis* such as hippopotamus (*Hippopotamus amphibious*), African elephant (*Loxodonta africana*) and African buffalo (*Syncerus caffer*) as well as small game animals such as bush pig (*Potamochoerus larvatus*) and duikers (*Sylvicapra* species) that are the preferred hosts of *G. austeni* [5,46,47]. In general, these protected areas are characterized by denser vegetation and lusher tree cover as compared to the communal farming areas.

Notwithstanding the apparent host preferences of these two species, both will feed on cattle [5] and cattle will probably be able to sustain tsetse populations in the absence of game in areas with suitable vegetation and climatic conditions. In the current study, cattle density contributed the highest percentage towards model fitting for both *G. austeni* and *G. brevipalpis*. For both species, cattle densities, irrespective of suitable habitat, showed a negative association with fly occurrence. This apparent contradictive observation may be related to cattle being used as live bait for tsetse control in north-eastern KZN. In the past the existing extensive dipping network, that was established mainly for tick control, was modified after an outbreak of nagana to include tsetse fly control by replacing the acaricide dipping agent with an insecticide [26,27]. Once the outbreak was under control the dipping agent was changed back to the acaricide to prevent ticks from developing insecticide resistance. The current live-bait control program in north-eastern KZN started in 2015 (personal communication Dr. L. Ntantiso, KZN Department of Agriculture and Rural Development) and the impact of this control method on

tsetse abundance is reflected by the predicted low suitability at diptanks in the communal farming areas where control is currently implemented.

The current control strategy in north-eastern KZN may have facilitated the creation of tsetse refuges in the protected areas. The perceived positive relationship between protected areas and tsetse occurrence and relative abundance were confirmed by the MaxEnt model. And this emphasizes a strong reinvasion potential from the uncontrolled protected areas into the communal farming areas. Over the last decade game farms, private nature reserves and other forms of wildlife-oriented land use, have increasingly become prominent features in KZN [48]. This expansion of protected areas combined with the absence or at least a general reluctance to suppress tsetse populations in these protected areas [49], increases the probability of migration into the farming areas with the associated constant threat of nagana transmission to livestock. This threat is exacerbated by the presence of buffalo and other wildlife hosts that are considered a reservoir host of nagana in some of these protected areas [12].

Previous probability of presence models of *G. austeni* and *G. brevipalpis* [20] looked at the estimated area covered by each species in north-eastern KZN at three probability threshold cut-offs, $P > 0.5$, $P > 0.25$ and $P > 0.125$. The total potentially infested area increased two-fold when the threshold was decreased from $P > 0.5$ to $P > 0.125$, from 5600 km$^2$ to 11 750 km$^2$ [20]. The current developed model included cattle density, human population, agricultural intensity and detailed vegetation biomes and the prediction area was extended to include Eswatini and southern Mozambique. The estimated area covered by both species at a probability threshold of $P > 0.5$ was 1700 km$^2$, 57.4 km$^2$ and 3901 km$^2$ for north-eastern KZN, Eswatini and MP, respectively. Therefore, the current model predicted a smaller tsetse fly-infested area than previous models. The reason for this is not clear but it can be speculated that this may be due to model refinement, changes in the habitat (e.g., bush clearing for agricultural purposes) as well as the apparent success of the current tsetse control actions in the area. Considering the sensitivity of tsetse to environmental factors this can even be indictive of the effects of climate change in the area. A more intensive study will be needed to pinpoint the exact reason for the predicted smaller tsetse fly-infested area.

The generated prediction model has been expanded to include the Gaza and Inhambane Provinces in Mozambique, an area suspected to be free of tsetse following a reduction in cattle and wildlife numbers due to human settlement and population expansion. This assumed tsetse-free area extends northwards for approximately 500 km from the northern part of the Matutuine tsetse belt up to the great Central and North tsetse belts that start at the Save River [9,11]. In agreement with historical data [10] and in view of a reduction in cattle and wild host populations in the area, the present model predicted very few areas with suitable habitat in the Gaza and Inhambane provinces of Mozambique (Fig 7). Suitable areas, with a low habitat suitability index, for *G. brevipalpis* (<0.6) and *G. austeni* (<0.2) were restricted to the coast (Fig 7). Factors that limit the presence of tsetse flies in the area may be the lack of suitable vegetation combined with a greater variation in temperature range in the interior. Historical collection data from 1984 for *G. austeni*, indicated that they were present 60 km west of Maputo and found to be present in the *Androstachys* forests [50,51].

The exact distribution of *G. austeni* and *G. brevipalpis* in Mozambique is not known, but it is suspected to be discontinuous due to natural barriers and land use changes. No tsetse control programmes have been implemented since the late 1960s and only limited surveys have been implemented in the last 20 years [6,9,11,52]. This emphasises the importance of the developed MaxEnt model as it provides indications of the most suitable areas that can be prioritised for the deployment of traps during surveys. This will significantly reduce time and cost of future surveys in this area.

Assessment of the MaxEnt model quality using an independent data set attained good performance with a high predictive power (AUC > 0.80 for both species). The available trap data however only partly validated this prediction model, as the present model seems to indicate a potential wider distribution of the two tsetse species as compared to the survey data. This underpins the importance of these models as tools not only for the planning of the surveys and the monitoring activities, but also for the suppression and later sterile male release activities.

The factors that contributed or shaped the distribution of *G. austeni* and *G. brevipalpis* as revealed by the present prediction model, indicated that changes in the climate, agricultural practises and land use can have a significant and rapid impact on tsetse presence/absence and abundance. The current trypanosomosis control strategy in South Africa (dipping with insecticides) can only be effective if a large proportion of the tsetse populations feed on cattle [28]. The observed trend in increased wildlife areas in KZN will lead to a decrease in the proportion of tsetse flies taking a blood meal from treated cattle. To manage trypanosomosis it will be crucial to supersede the current South African control strategy of using cattle as live baits with a more sustainable AW-IPM strategy. In 2007, an AW-IPM strategy that include an SIT component was proposed to establish a tsetse fly free South Africa [7]. The proposed AW-IPM strategy suggested the division of the infested area into four zones from south to north with the successive implementation of four phases (pre-suppression, suppression, SIT and post-eradication) in each zone following the rolling carpet principle [7]. In respecting the principles of an AW-IPM approach, the control effort should be directed against the entire insect population, and therefore, the proposed AW-IPM strategy of 2007 should be modified to include the tsetse populations from southern Mozambique and Eswatini.

## Supporting information

**S1 Fig. Uncertainty grid for the habitat suitability index model for *Glossina austeni* and *Glossina brevipalpis* (https://dataverse.harvard.edu/dataset.xhtml?persistentId=doi:10.7910/DVN/PA7U7L).**
(TIF)

## Acknowledgments

We thank our co-workers in the Epidemiology, Parasites & Vectors Programme as well as the staff at the KwaZulu-Natal Tsetse Research Station for their valuable assistance.

## Author Contributions

**Conceptualization:** Chantel J. de Beer, Jérémy Bouyer, Marc J. B. Vreysen, Gert J. Venter.

**Data curation:** Moeti O. Taioe, Fernando C. Mulandane, Luis Neves, Sihle Mdluli.

**Formal analysis:** Chantel J. de Beer, Ahmadou H. Dicko, Laure Guerrini, Jérémy Bouyer, Marc J. B. Vreysen, Gert J. Venter.

**Funding acquisition:** Chantel J. de Beer, Marc J. B. Vreysen, Gert J. Venter.

**Investigation:** Chantel J. de Beer, Jerome Ntshangase, Percy Moyaba, Fernando C. Mulandane, Sihle Mdluli, Laure Guerrini.

**Methodology:** Chantel J. de Beer, Ahmadou H. Dicko.

**Project administration:** Chantel J. de Beer.

**Resources:** Chantel J. de Beer.

**Supervision:** Luis Neves, Jérémy Bouyer, Marc J. B. Vreysen, Gert J. Venter.

**Visualization:** Chantel J. de Beer, Ahmadou H. Dicko.

**Writing – original draft:** Chantel J. de Beer, Ahmadou H. Dicko, Jérémy Bouyer, Marc J. B. Vreysen, Gert J. Venter.

**Writing – review & editing:** Chantel J. de Beer, Ahmadou H. Dicko, Jerome Ntshangase, Percy Moyaba, Moeti O. Taioe, Fernando C. Mulandane, Luis Neves, Sihle Mdluli, Laure Guerrini, Jérémy Bouyer, Marc J. B. Vreysen, Gert J. Venter.

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
