## [Decision Letter · Decision Letter 0]

26 Jul 2021

Dear Dr De Beer,

Thank you very much for submitting your manuscript "A distribution model for Glossina brevipalpis and Glossina austeni in Southern Mozambique, Eswatini and South Africa for enhanced area-wide integrated pest management approaches" for consideration at PLOS Neglected Tropical Diseases. As with all papers reviewed by the journal, your manuscript was reviewed by members of the editorial board and by several independent reviewers. In light of the reviews (below this email), we would like to invite the resubmission of a significantly-revised version that takes into account the reviewers' comments. 

We cannot make any decision about publication until we have seen the revised manuscript and your response to the reviewers' comments. Your revised manuscript is also likely to be sent to reviewers for further evaluation.

Sincerely,

Marc Choisy

Associate Editor

Anthony Papenfuss

Deputy Editor

Reviewer's Responses to Questions

**Key Review Criteria Required for Acceptance?**

**Methods**

-Are the objectives of the study clearly articulated with a clear testable hypothesis stated?

-Is the study design appropriate to address the stated objectives?

-Is the population clearly described and appropriate for the hypothesis being tested?

-Is the sample size sufficient to ensure adequate power to address the hypothesis being tested?

-Were correct statistical analysis used to support conclusions?

-Are there concerns about ethical or regulatory requirements being met?

Reviewer #1: The methods are clearly indicated and the results reflect this.

Reviewer #2: There is insufficient detail provided on a number of aspects of the methodology. 

- The objective of the study should be stated more clearly at the end of the introduction.

Study site descriptions and tsetse data collection section:

- Have any of these data been published before? Reference if so. 

- How were the locations of tsetse trapping sites chosen given this is important for MaxEnt to produce robust results?

- Line 196-197 Does this mean that some traps were in the same location for 2313 days? Did this affect trap catches, i.e. you may expect to see a decline in population over time? 

- It isn’t stated what type of data were collected on the tsetse trapped. Presumably they were identified morphologically, sexed and counted?

Analyses:

- Line 228 ‘ANOVA was used to differentiate between the mean tsetse fly AD’. Do you mean it was used to compare AD between different locations? How were the confidence intervals calculated on the mean ADs?

- Line 229 Apparent density needs to be defined, and it should be explained how it was calculated

- From line 233 The process for generating pseudoabsences isn’t very clear. How exactly were they generated and how many were used? 

- From line 233 How were repeat samples from the same location used in the analysis? The filter process isn’t explained very clearly – were repeat measures subsumed into a single presence point by this filtering? Was the season of tsetse collection considered in the analysis?

- Line 253 – Field surveys are mentioned here but not described. What field data were collected and how were the data used?

- Line 302 What exactly is meant by ‘to assess the effect of pseudo-absence’ 

- Line 303 Two independent datasets are mentioned here but not described. In the abstract one independent dataset is mentioned. Details of these datasets should be included. 

- The method used to measure uncertainty is not described.

Reviewer #3: yes to all questions

**Results**

-Does the analysis presented match the analysis plan?

-Are the results clearly and completely presented?

-Are the figures (Tables, Images) of sufficient quality for clarity?

Reviewer #1: The results are adequately presented and all figures are also of sufficient quality.

Reviewer #2: Mostly the results are clearly described although some things had not been included in the methods. Figures are clear and nicely presented.

Line 359-362 This finding regarding which factors influence distribution is obviously important. Whilst it is clearly stated here that human and cattle density have a negative association with occurrence, in other places e.g. line 440, 477, abstract, it isn’t made clear that it is a negative relationship. 

Lines 417-421 Is extrapolating beyond the limits of data collection justified? This part should also be described in the methods.

Reviewer #3: yes to all

**Conclusions**

-Are the conclusions supported by the data presented?

-Are the limitations of analysis clearly described?

-Do the authors discuss how these data can be helpful to advance our understanding of the topic under study?

-Is public health relevance addressed?

Reviewer #1: The results are also adequately discussed.

Reviewer #2: In general the relevance of the findings is discussed and mostly valid conclusions are reached. The limitations of the approach chosen are not addressed sufficiently. 

Specific comments: 

 - Line 436 ‘whilst incorporating the ecology of both species’ I’m not convinced this approach does incorporate the ecology – remove or be more specific

 - Line 440 should say presence of cattle, not presence of host animals, since you didn’t assess wildlife hosts. 

 - Lines 476-486 The logic isn’t totally clear here but I think you are suggesting that the negative relationship between cattle density and tsetse presence is most likely due to dipping of cattle in farming areas. Whilst I agree it is plausible, how can you differentiate this hypothesis from there being less appropriate tsetse habitat in farming areas? 

 - Line 496 It would be more accurate to say the presence of buffalo and other wildlife hosts, since there are many species that can carry T. congolense and/or T. vivax.

Reviewer #3: yes to all

**Editorial and Data Presentation Modifications?**

Reviewer #1: Minor Revision

Reviewer #2: - Line 51 should be ‘practices’

- Reference 14 does not support the statement where it is used, since it doesn’t include any assessment of vector competence. Reference 15 is appropriate but the sentence should read ‘more competent for T. congolense’, since it was only this species that was assessed. 

- Line 95-96 check grammar

- What is already known about the habitat preferences or predictors of brevipalpis and austeni presence should be stated in the intro. There is literature on this that is cited later, e.g. ref 18, 28, 39 but this should be mentioned in the introduction and the specific novelty of this study identified.

- Lines 128-129 The Leak book, in its entirety, is an odd reference to use here. The more recent and specific literature on this topic should be cited, e.g. Hargrove et al. 2012. Also applies to line 541

- Lines 130-133 States that an AW-IPM strategy that includes an SIT component is required. Whilst an AW-IPW is clearly required, SIT is not the only option and the wording of this sentence should be edited to reflect that. 

- Line 133-135 Reference 22 should come at the end of the first part of the sentence. 

- Line 153-155 This sentence is rather vague and it isn’t clear whether it is stating an objective of the study or simply commenting on the potential use of the distribution maps in the future. Suggest remove and instead add a sentence clearly stating the objective of this study. 

- Line 193 – does this mean there were seven sachets per trap?

- Line 260-261 Gridded livestock data – the version should be stated and appropriate reference included. World Database on Protected Areas should be referenced appropriately. 

- Lines 316 to 353 I wonder if this whole section would be better presented in a table or a figure.

Reviewer #3: (No Response)

**Summary and General Comments**

Reviewer #1: This is a technical report about “A distribution model for Glossina brevipalpis and Glossina austeni in Southern Mozambique, Eswatini and South Africa for enhanced area-wide integrated pest management approaches”. 

The paper is clear and very well written, and gives insight into the importance of the development of habitat suitability maps, and an assessment of the factors that regulate the presence or absence of the targeted species for the successful implementation of an AW-IPM programme, especially those that include an SIT component. The materials and methods are clearly indicated and the results reflect this. The results are also adequately discussed.

It is recommended that the manuscript be accepted for publication.

Some minor errors and modifications are suggested here below.

Materials and methods

Line 165: add “,” after “In Eswatini”

Lines 170 – 172: The fig 1 shows tsetse apparent densities and could be announced in the results section. May be it will be better to designed a figure which showing only the study area without the tsetse densities.

Line 190: “are” must be added after “incorporated”?

Lines 196 – 199: the authors indicated that “In total 160 H-traps were deployed for a minimum of 19 and a maximum of 2313 trapping days. Flies were collected from the traps and the traps serviced (clearing of vegetation, odour replacement, replace the traps when the colours are faded etc.) every 14 days.”

Generally, for just tsetse sampling for presence/absence and abundance evaluation, around 3 days of traps deployment is enough but in this study, the trap was kept until 2313 trapping days. It seem to be a tsetse control program where traps were deployed for monitoring, the authors must notify that in the methodology section.

Line 229: what means “AD”? …… add “apparent density (AD)”

Line 231: add “is” before “< 0.05”

Line 299: add “,” after “derived index”

Results

Line 317: put space after “±” in the sentence “(2.14 ±5.4) in MP. However, the lower mean AD (0.08 ±0.2)”. The same correction is necessary in the rest of the result section.

Line 321: delete the space between “(�x” in the sentence “followed by shrub savannah (�x AD of 6.46 ±7.9)”. The same correction is necessary in the rest of the result section.

Line 336: add “,” after “In Eswatini”

Discussion

Line 544: add “,” after “In 2007”

References

Line 600: replace “van” by “Van”

Reviewer #2: This paper presents a distribution model for G. austeni and G. brevipalpis for the tsetse belt that covers parts of South Africa, Mozambique and Eswatini. The novelty of the work is that it uses data collected in all three countries to look at area-wide distribution to assist development of regional tsetse control plans and will provide a valuable tool for this purpose.

Weaknesses – The study does not address a new question or describe a novel approach, although it is a useful example of using regional distribution modelling to inform control. The MaxEnt approach used, whilst commonly used for predicting the probability of presence of vectors, uses presence only data inputs. Other approaches could have been considered to make the most of the abundance data collected. For several aspects of the methodology, the level of detail provided is not sufficient to fully understand what has been done (see specific comments). The limitations of the approach are not discussed. The clarity of writing could be improved in places (see specific comments).

Reviewer #3: Comments on PNTD-D-21-00569

This paper uses entomological data on captures of the two main tsetse vectors of african trypanosomes in the southern Africa region, in order to develop prediction maps and identify habitat suitability for tsetse presence/absence. These information are key, if not essential to the development of tsetse control programmes. The ms is of very good quality, and is of help and relevance.

Majors comments

The authors should add information on the contraint due to AAT which is absent here : what are the prevalences ? costs associated ? etc., any information would help. The reader understands that the initiative comes from South Africa where there are control programmes on-going, but how is it preceived in Mozambique and Eswatini ? I appreciate this goes a bit beyond the purely scientific aspcet of the paper, but it would certainly help also. 

Minor comments

Introduction, line 76 : « and they infest about 10 million km² in sub-Saharan Africa » : do you really believe this ? ref 2 is not appropriate here, and may be certainly cited elsewhere in the text. But you may rather write « are reported to infest », or include some nuance here, as you are well placed to know what you are speaking about…

Introduction, line 79-80 : please include a ref for HAT, since you mention two diseases, but only give ref for the animal one. In addition, speaking about the human disease, you may include an impact more related to health than to economy…

Introduction, line 89 : having written Trypanosoma in full a first time here, you may replace it by « T. » on all other occurrences (l. 89, 90, 92 ? etc).

Introduction, lines 130-132 : « The establishment of a tsetse-free zone in southern Africa would be a more sustainable solution to the nagana problem, but would require an AW-IPM strategy that includes an SIT component ». Although the beginning of your statement is correct, please end it by something less subjective, and more appropriate. I don’t think SIT is the only tool that can lead to tsetse eradication, especially in this southern area of tsetse distribution where tsetse have also been eradicated from neighbouring countries by aerial spraying, correct me if I am wrong ? 

Discussion, lines 428-430 : « A sound understanding of the potential distribution of targeted tsetse fly species will be essential for the successful implementation of an AW-IPM programme, especially those that include an SIT component ». Not only I agree here, but please be more global because this « sound understanding » is essential, if not mandatory, for ANY control programme, again yes for those which include an SIT component, but not only for them…

Discussion lines 506-508 : « Therefore, the current model predicted a smaller tsetse fly infested area than previous models. This may be due to model refinement, changes in the

habitat as well as the apparent success of the current tsetse control actions in the area. » may be you could expand a bit here, to help the reader’s understanding. Why those « changes in habitat », what would be the cause, e.g. human growth, habitat destruction, why not even climate change, I am sure you know better than me.

Discussion line 510-511 : « an area suspected to be free of tsetse following a reduction in cattle and wildlife numbers », the reader may want to understand why is that so ? I mean why a reduction in cattle and wildlife numbers?

PLOS authors have the option to publish the peer review history of their article (what does this mean?). If published, this will include your full peer review and any attached files.

Reviewer #1: No

Reviewer #2: No

Reviewer #3: No
---

## [Editor Report · Decision Letter 1]

9 Nov 2021

Dear Dr De Beer,

We are pleased to inform you that your manuscript 'A distribution model for Glossina brevipalpis and Glossina austeni in Southern Mozambique, Eswatini and South Africa for enhanced area-wide integrated pest management approaches' has been provisionally accepted for publication in PLOS Neglected Tropical Diseases.

Best regards,

Marc Choisy

Associate Editor

Anthony Papenfuss

Deputy Editor

---

## [Editor Report · Acceptance letter]

20 Nov 2021

Dear Dr de Beer,

We are delighted to inform you that your manuscript, "A distribution model for Glossina brevipalpis and Glossina austeni in Southern Mozambique, Eswatini and South Africa for enhanced area-wide integrated pest management approaches," has been formally accepted for publication in PLOS Neglected Tropical Diseases.

Best regards,

Shaden Kamhawi

co-Editor-in-Chief

Paul Brindley

co-Editor-in-Chief
